# A Study on CP-ABE-Based Medical Data Sharing System with Key Abuse Prevention and Verifiable Outsourcing in the IoMT Environment

**DOI:** 10.3390/s20174934

**Published:** 2020-08-31

**Authors:** Yong-Woon Hwang, Im-Yeong Lee

**Affiliations:** Department of Computer Science and Engineering, Soonchunhyang University, Asan 31538, Korea; hyw0123@sch.ac.kr

**Keywords:** internet of medical things, cloud storage, access control, CP-ABE, traceability, verifiable outsourcing, user privacy protection

## Abstract

Recent developments in cloud computing allow data to be securely shared between users. This can be used to improve the quality of life of patients and medical staff in the Internet of Medical Things (IoMT) environment. However, in the IoMT cloud environment, there are various security threats to the patient’s medical data. As a result, security features such as encryption of collected data and access control by legitimate users are essential. Many studies have been conducted on access control techniques using ciphertext-policy attribute-based encryption (CP-ABE), a form of attribute-based encryption, among various security technologies and studies are underway to apply them to the medical field. However, several problems persist. First, as the secret key does not identify the user, the user may maliciously distribute the secret key and such users cannot be tracked. Second, Attribute-Based Encryption (ABE) increases the size of the ciphertext depending on the number of attributes specified. This wastes cloud storage, and computational times are high when users decrypt. Such users must employ outsourcing servers. Third, a verification process is needed to prove that the results computed on the outsourcing server are properly computed. This paper focuses on the IoMT environment for a study of a CP-ABE-based medical data sharing system with key abuse prevention and verifiable outsourcing in a cloud environment. The proposed scheme can protect the privacy of user data stored in a cloud environment in the IoMT field, and if there is a problem with the secret key delegated by the user, it can trace a user who first delegated the key. This can prevent the key abuse problem. In addition, this scheme reduces the user’s burden when decoding ciphertext and calculates accurate results through a server that supports constant-sized ciphertext output and verifiable outsourcing technology. The goal of this paper is to propose a system that enables patients and medical staff to share medical data safely and efficiently in an IoMT environment.

## 1. Introduction

Cloud computing has become very popular, and users can safely store and share data via the cloud. With these advantages, it is possible to apply cloud environments to many fields, including Business to Consumer (B2C), enterprise, public, and Internet of Things (IoT) services. Recently, IoT and the cloud have been combined to collect patient data as via wearable devices in the medical field through Internet of Medical Things (IoMT), and to store it in a hospital cloud environment so doctors and nurses can remotely monitor it to check the patients’ health. Figure 1 shows a flow diagram of a medical data sharing system for patients in a cloud-based IoMT environment. Here, the data owner is assumed to be a patient, and the cloud server is assumed to be a medical environment server. A data user refers to a user who has the authority to check the data of the data owner, such as a doctor or a nurse, regardless of gender. For example, it means a doctor or nurse in charge who is examining a patient. As shown in Figure 1, the patient’s medical data (physical signal) collected by healthcare wearables is stored on a smartphone, and if it is transmitted to the hospital cloud server, a doctor or nurse in the department in charge of the patient can remotely monitor the patient’s data. By examining the patient’s medical data and sending prescription data to the cloud server, the patient can check the prescription results smart devices. Therefore, since the introduction of IoMT enables safe data sharing between patients and doctors or nurses in hospitals, patients can manage their health in a fast and convenient environment. However, there are some security threats when sharing data in an IoMT cloud. First, service providers cannot be completely trusted. If a user employs a cloud provided by a large enterprise, stored data are safe from external threats, but the cloud provider can infringe user privacy because the provider has access to the data. Especially in the medical cloud environment, the stored data is very sensitive. Therefore, security of stored data is important in a cloud environment. In addition, access control technology and user authentication technology for accessing encrypted data are required. Various security technologies exist, among which Attribute-Based Encryption (ABE) is popular. ABE has the advantage of ensuring the user’s anonymity because the data is encrypted with the user’s attributes. This is used to protect privacy in an environment where sensitive data such as healthcare information is sent and received. ABE includes both ciphertext-policy (CP)-ABE and key-policy (KP)-ABE. The differences are described in Section 2 and the proposed method considers the CP-ABE method [1,2]. The CP-ABE method is an encryption that allows 1:N data sharing. When a data owner encrypts and uploads data, any user meeting owner-defined attributes can decrypt the ciphertext (CT). However, an attribute authority (AA) is required to manage user attributes. Nevertheless, CP-ABE-based access control techniques are often used to access data in cloud environments [3,4]. Various CP-ABE based access control technologies have been studied to date. The existing CP-ABE access control methods do not identify the user who issues the secret key. Accordingly, a user can leak the secret key for his own benefit and that user cannot be traced. For example, in Figure 1, if a doctor gives another user the secret key to access the medical cloud for malicious purposes, the other user can access the medical cloud with that key. If the secret key sent to another person is misused and you want to track the user (doctor) who issued that secret key, the secret key does not reveal the user who was issued the secret key because there is no information about the user [5]. This is a situation that can occur due to the use of CP-ABE. To solve this problem, when the attribute authority, which verifies the attribute and issues the key, issues the key for each user, it records the user’s ID value or user signature value in the key parameter [6,7,8,9,10,11]. If the key is misused after the key is delegated or delivered to someone, research is being conducted to verify the identity by tracking the user who initially issued the key with the user ID value or user signature value included in the key parameter. However, since user attributes and information are exposed to Attribute Authority (AA), it may infringe on the privacy of users. Most of all, attribute-based encryption is characterized by not having to reveal the identity of multiple participants because it accesses the user’s attributes. Therefore, if the user ID value is managed by the AA, the anonymity of the cloud user will not be ensured. Another problem is that because data is encrypted with attribute-based encryption, the size of the ciphertext increases with the number of attributes contained in the access structure [12,13,14]. This can waste space in cloud storage, and when the user decrypts the ciphertext the amount of computation increases with the number of attributes, which affects the user’s ciphertext decryption speed. Lastly, since the amount of computation is large when the user decrypts the ciphertext, an outsourcing server is needed to support the decryption operation during distribution. Additionally, the user should know whether the result value calculated by the outsourcing server is a properly calculated result value according to the result value after the final decryption. Therefore, a verification process is needed to prove that the result calculated by the outsourcing server is a properly calculated result [15,16,17,18,19,20,21,22,23]. In this paper, we propose a system that can safely share medical data among multiple users who use the cloud in an IoMT environment and provide protection for user privacy and from key abuse. In detail, based on CP-ABE access control techniques where various security requirements have been studied, such as outsourcing operations, partial decoding, and ciphertext constant-size output, we will add techniques that can track the identity of the user issued the key, as well as verification outsourcing techniques.

The contributions in this paper are as follows. (1) User anonymous ID usage and traceability. Users use anonymous IDs in the cloud environment and there is an object called Trace Authority (TA) in the cloud to track down and identify the user who has been issued the key in the event of a problem with a user’s key [24]. (2) Output of constant size of ciphertext, regardless of the number of attributes. The efficiency of the storage space of the cloud storage can be improved. (3) Provide verifiable outsourcing techniques. The Access Control (AC) server can reduce the ciphertext decryption computations for user decryption by performing partial decryption on outsourcing servers. In addition, verification allows users who have performed final decryption to verify the integrity of outsourced ciphertext transformations and whether the data was uploaded by the data owner.

The scheme proposed in this paper is not intended for all cloud environments. It will be applicable to cloud environments focused on security of sensitive data such as military and medical and protects the privacy of users accessing the data. Each chapter of this paper is as follows. Section 2 is a related study that describes the formulas used for attribute-based encryption (bilinear map and complexity assumptions), data security technologies in cloud environments, and content on attribute-based encryption (definition, key abuse problem, need for verifiable outsourcing technology, previously proposed studies), and security models. Section 3 describes the security requirements in the IoMT environment, and Section 4 describes the proposed scheme. Section 5 explains the analysis of the security and efficiency of the proposed scheme, and Section 6 concludes with a conclusion.

## 2. Related Works

This section describes bilinear mapping and attribute-based encryption, and explores the need for traceability and security models in the existing CP-ABE methods.

### 2.1. Bilinear Map

Consider an additive group $G_{1}$ and a multiplicative group $G_{2}$ having the same constant q. Assuming that it is difficult to solve the discrete logarithm problem in groups, let P be the constructor of the additive group $G_{1}$. Let $e:G_{1}\times G_{1}\to G_{2}$ be a folded bilinear map satisfying the following properties:Bilinearity: For all P,Q∈1 and all a,b∈Z, e(aP,bQ)=e(P,Q)ab.Non-Degeneracy: For all $Q\in G_{1}$ if, e(P,Q)=1, then P=0.Computability: There exists an efficient algorithm computing e(P,Q) for all $P,Q\in G_{1}$.

### 2.2. Complexity Assumption

#### 2.2.1. Bilinear Diffie Hellman (BDH) Assumption

Given two pairs $(g^{a},\;g^{b},\;g^{c},\;T=e{\left(g,g\right)}^{abc}$) and $(g^{a},\;g^{b},\;g^{c},\;W=e{\left(g,g\right)}^{z}$), the deterministic BDH assumption means that there is no algorithm A that can distinguish the two pairs with a meaningful probability. Here it is $a,\;b,\;c,\;z\in Z_{p}.$ If algorithm A, which solves the deterministic BDH assumption, satisfies $\left|\mathrm{Pr}\left[A\left(g^{a},\;g^{b},\;g^{c},\;T\right)=1\right]-\mathrm{Pr}\left[A\left(g^{a},\;g^{b},\;g^{c},\;W\right)=1\right]\right|\geq \epsilon$, it is said that algorithm A has a profit of ϵ.

#### 2.2.2. Bilinear Diffie Hellman Exponent (BDHE) Assumption

The deterministic BDHE assumption means that, given $\left(h,\;g,\;g^{\alpha }\ldots .,g^{\alpha }{}^{\beta },\;g^{\alpha }{}^{\beta +2},\ldots g^{\alpha }{}^{2\beta }\right)$, there is no algorithm A that can compute $T=e{\left(h,g\right)}^{\alpha }{}^{\beta +1}$ with a meaningful probability. Here is $h,g\in G_{1}$. As defined by $g_{i}=g^{\alpha^{i}}\left(i=1,\ldots ,2B\right)$ and $g_{\alpha ,\beta }=\left(g_{1},\ldots ,g_{B},\;g_{B+2},\;\ldots ,\;g_{2B}\right)$, when the next two pairs are ($h,g,\;g_{\alpha ,\beta },\;T=e{\left(h,g\right)}^{\alpha }{}^{\beta +1}$), ($h,g,\;g_{\alpha ,\beta },\;W=e{\left(h,g\right)}^{z}$), the algorithm A solving the deterministic BDHE assumption is $\left|\mathrm{Pr}\left[A\left(h,g,\;g_{\alpha ,\beta },\;T\right)=1\right]-\mathrm{Pr}\left[A\left(h,g,\;g_{\alpha ,\beta },\;W\right)=1\right]\right|\geq \epsilon$ is satisfied, algorithm A is said to have profit of ϵ.

### 2.3. Data Security Technology in the Cloud

Encryption technologies protecting stored cloud data include symmetric and public key encryption. ID-based encryption (IDE) (including public key encryption, ABE, and proxy re-encryption) is used to safely share data. Existing symmetric encryption that uses the same key for encryption and decryption in large distributed environments, including the cloud environment, has key distribution and management problems. On the other hand, the existing asymmetric encryption methods using a public key for encryption and a private key for decryption lack computational efficiency. It is not recommended to use such a 1:N environment to encrypt data before sending data to a group because the data owner needs to know who the recipients are, requiring IDs or public keys before encrypting data and sending them individually to authorized users. Several cryptographic access control methods have been developed to reduce the computational costs of existing operations and achieve data confidentiality, access control, and representational policies for data stored in dynamic environments. The most frequently used access control type is ABE [3]. 

Attribute-based encryption is encryption that allows 1:N different recipients, and when the data owner encrypts the data and uploads it to storage, an unspecified number of participants with the corresponding attributes can access the ciphertext and decrypt it. In other words, the subject who accesses the data ciphertext can designate the owner of the data based on the user’s attributes. Accordingly, data access control and data sharing system methods using attribute-based encryption in the cloud environment are being studied a lot. This study also intends to propose a secure data sharing system using a CP-ABE type of attribute-based encryption.

### 2.4. Attribute-Based Encryption (ABE)

ABE performs encryption and decryption depending on user attributes and an access structure. ABE includes CP-ABE and KP-ABE. At the left of Figure 2 is the CP-ABE method [1,2]. The owner of the data creates an access structure with the attributes of the users who can access the data and creates a ciphertext using the public parameters, master key, and access structure. After, the data is encrypted, and it is transmitted to the cloud. Thereafter, users who want the data then access the cloud to receive the ciphertext, and compare their own attributes with those specified in the ciphertext’s access structure to decrypt the ciphertext. For example, if a user who wants to access the data has the [Internal Medicine] and [Doctor] attributes, as shown in Figure 2, the data owner creates and encrypts an access structure including the [Internal Medicine, Doctor] attributes. Then, only the users with the attributes matching the access structure can decrypt the ciphertext. 

The KP-ABE method is as shown at the right of Figure 2. When the data owner creates a ciphertext, the data is encrypted with the attributes of the users who can access the data and is transmitted to the cloud. After that, a user who wants the data can access the cloud to receive the ciphertext, and the user can decrypt the ciphertext by creating an access structure based on their attributes and by generating a key to decrypt the ciphertext. For example, the data owner encrypts the data with the property [Internal Medicine, Nurse] and uploads it to the cloud. After that, a user with the attributes of [Internal Medicine] and [Nurse] can create an access structure to create a key to decrypt the ciphertext. 

The difference is between who generates the access structure and key. The owner of the data can specify who can access the data in both CP-ABE and KP-ABE. CP-ABE requires an authority that can issue a key after verifying user attributes, but KP-ABE does not need an authority that can issue a key because the user generates a key directly. This paper projects conducted research on CP-ABE used in a common cloud sharing environment. 

Figure 3 is a schematic diagram of a CP-ABE-based data sharing system for safe and efficient data sharing in a medical cloud environment. The data owner creates an access structure with the attributes of the users who can access the data and uploads it to the cloud. Later, any number of unspecified users with the corresponding attributes may access and decrypt the data. The advantage here is that the data can be accessed and decrypted when conditions such as attribute matching are met, so that anonymity for identities of users is provided. In addition, compared to the proxy re-encryption or ID-based encryption methods, it has an advantage that the CP-ABE method does not require a verification process, whereas the user receives data from the data owner for confirmation of data access when the user requests data from the cloud server. In addition, it can be applied to data sharing environments of IoMT and military and corporate cloud systems [5].

#### 2.4.1. Key Abuse Issues in CP-ABE

In existing CP-ABE methods, a user requests registration from the AA, and the AA generates and transmits a key employing user attributes to permit decryption. In this case, the secret key $\mathrm{SK}=\left(D=g^{\frac{\alpha +\gamma }{\beta }},\;\forall j\in S:D_{j}=g^{\gamma }\cdot H{\left(j\right)}^{r_{j}},D_{j}{}^{\prime }=g^{r_{j}}\right)$ is a parameter operation that includes the user attribute value, and there is no value to identify the user. Thus, if the user maliciously distributes the key to a third party, that party can access the cloud and decrypt the data. Figure 4 shows a scenario where a package file worth $100 is placed for 10 days on the cloud server. User 1 purchases it and employs it for 10 days. If User 1 sells the key to Users 2 and 3 for $60 each, those users can also access the file and user 1 cannot be identified. The problem here is that there is a process in which the key is distributed, but above all, the problem is that the user who was issued the key cannot be known. This is a fundamental problem with CP-ABE. To solve this, as shown in Table 1, CP-ABE schemes that provide traceability to prevent key abuse have been proposed [5,6,7,8,9,10]. However, CP-ABE schemes that provide traceability can invade user privacy by exposing the users’ identifier values to the AA when the keys are issued, and thereby the anonymity of the users is reduced. Accordingly, this study proceeds in the direction of verifying the identity of the user through a tracking process when the key that is distributed is later misused while protecting the privacy of the users through anonymous IDs.

#### 2.4.2. Necessity of Verifiable Outsourcing Techniques

As the cloud features many outsourced servers processing data and owner/user messages, computational demands are high. In addition, data can be shared among mobile devices and the Internet of Things. However, users do not know if data was properly converted by the process of encrypting and decrypting messages on a server that supports outsourcing. In particular, although the data owner encrypts the message M, if the value decrypted by the actual user is M′, it is impossible for the user to know that the M′ value is different from the original M. Accordingly, it is necessary for the user to verify that the calculated value returned by the outsourcing server is a properly calculated value while performing the final decryption [15,16,17,18].

#### 2.4.3. Previously Proposed CP-ABE Scheme

To date, CP-ABE aims to improve the efficiency of computation by preventing key abuse and decrypting ciphertext, as shown in Table 1. Table shows CP-ABE methods that deal with key security as well as other security requirements for CP-ABE methods, such as traceability for tracking key abuse, constant size output of ciphertext, and the application of outsourcing techniques for efficiency of operation. Among the methods to provide traceability, the Liu scheme [6], Yu scheme [8], and Li scheme [5] are $SK=\left(D=g^{\frac{\alpha +\gamma }{\beta }},\;\forall j\in S:D_{j}=g^{\gamma }\cdot H{\left(j\right)}^{r_{j}},D_{j}{}^{\prime }=g^{r_{j}}\right)\;\mathrm{and}$ contain the user value (signature, ID) value. AA is the identity table that manages the user’s information, and the issued key suggests a scheme. If the issued key is misused, a value that can obtain the user’s identifier is extracted from the misused key, and the value is sent to AA to verify the identity of the user who initially issued the key. Assuming that in Li’s scheme, AA maintains the ID table of registered users, AA tracks the user after verifying the key with Key Sanity Check as follows. The Key Sanity Check process in the Li scheme is as follows: (1)$$K^{\prime }\epsilon Z_{N},\;K,\;L,\;L^{\prime },\;K_{x}\epsilon \;G\;e\left(g,L^{\prime }\right)=\;e\left(g^{\alpha },L^{\prime }\right)\neq 1\;e\left(g^{\alpha }\cdot g^{K^{\prime }},K\right)=e{\left(g,g\right)}^{\alpha }\cdot e\left(L^{K^{\prime }}\cdot L^{\prime },h\right)\neq 1\;x\in S,s.t.e\left(U_{x},L^{K^{\prime }}\cdot L^{\prime }\right)=e{\left(g,K_{x}\right)}^{\alpha }\neq 1$$

After verification, $K^{\prime }=c$ is extracted from the key and transmitted to the AA, and the AA checks the user ID value corresponding to $K^{\prime }=c$ to confirm the identity of the user who was first issued the key.
(2)$$SK=\left(K=g^{\frac{\alpha }{a+C}}h^{t}R,\;K^{\prime }=c,L=g^{t}R_{0},L^{\prime }=g^{at}{R^{\prime }}_{0},\;{\left\{K_{x}=U_{x}{}^{\left(a+c\right)t}R_{x}\right\}}_{x\in S}\right)$$

However, if the AA, which is an organization that manages attributes included in the Li scheme, manages the identity table of registered users, the anonymity of users will not be preserved, which infringes the privacy of users. In addition, the Li scheme, which proposed traceability as the main core, was not considered in terms of efficiency. In particular, the size of the ciphertext is proportional to the number of attributes, which wastes space in the storage of data and increases the user’s decryption computations.

In 2015, the Hahn scheme [13] proposed an attribute-based, secure data-sharing technique supporting outsourcing of cryptographic and decryption operations of a constant size. However, the system is applicable only to private cloud environments. It is not suitable for public cloud environments that share data with others. In 2017, the Helil scheme [15] proposed a hierarchical attribute-based access control method using a constant size ciphertext, but the size of the ciphertext is proportional to the number of attributes, and storage space is wasted. Since the $\forall y\in \mathrm{AS}:\;C_{y}=g^{q_{y}\left(0\right)},\;C_{y}^{\prime }=H{\left(att\left(y\right)\right)}^{q_{y}\left(0\right)}$, part of the ciphertext (CT) assigned to the access structure, the key size, increases with the number of attributes:(3)$$\mathrm{CT}=\left(\begin{array}{c} AS,\;C^{\prime }={\left\{M\right\}}_{Key},C^{*}=g^{s},\;\forall y\in AS:\;\\ C_{y}=g^{q_{y}\left(0\right)},\;C_{y}^{\prime }=H{\left(att\left(y\right)\right)}^{q_{y}\left(0\right)} \end{array}\right)$$

Although the computations in the encryption process do not change by multiplying the number of different attributes by one constant size when creating the access structure, research on constant-size output is needed because the size of the ciphertext can be minimized. Table 2 shows the CP-ABE methods that dealt with the verifiable outsourcing method and the constant size output of the cipher text among the various security requirements for CP-ABE methods. 

The schemes (Lai scheme [16], Jiguo Li scheme [20], and Zhidan Li scheme [21]) that proposed verifiable outsourcing encrypt two different messages when generating a ciphertext and generate a value that can verify the two messages. Later, the user verifies the integrity of the message by creating a verification value with the two messages obtained by decrypting the ciphertext. For illustration, the Jiguo Li scheme [20] generates the ciphertext as follows.
(4)$$\mathrm{CT}=\left(AS,\;\hat{C},\;C_{1},C_{2},C_{3},{C_{1}}^{\prime },{C_{2}}^{\prime },{C_{3}}^{\prime }\right)$$
where $C_{1},C_{2},C_{3}$ is the encryption value for M, ${C_{1}}^{\prime },{C_{2}}^{\prime },C_{3}^{\prime }$ is the encryption value for $M^{\prime }$, and $\hat{C}=u^{H\left(M\right)}v^{H\left(M^{\prime }\right)}d$ is the step of verifying the message integrity with the message $M,\;M^{\prime }$ obtained by the user later. In the above method, the integrity of the message obtained by the user is verified. However, it is not known whether the data was actually uploaded by the owner of the data, and the encryption and decryption operations are high compared to the previous CP-ABE methods. Therefore, it is necessary to introduce a signature-based verification step that can verify whether the data was uploaded by the data owner while also verifying the integrity of the message acquired by the user, and it is necessary to introduce an outsourcing method capable of efficient calculation.

#### 2.4.4. Security Model

Our attack model is similar to that of the CP-ABE introduced by Zhou and Huang (2010), based on a semantic security game [25]. The game engages in message transmission and reception between probabilistic polynomial time adversaries (PPT attacker) and a challenger; the probability that the attacker can finally compromise the safety of the CP-ABE technique is the game. The probability of winning is derived. The game details follow:Init: The attacker selects the challenger access policy W and gives it to the challenger.Setup: The challenger performs the Setup step to generate the public key ${\mathrm{PK}}_{S}$, and the challenger sends ${\mathrm{PK}}_{S}$ to the attacker.Phase: The attacker requests the secret key for the following access policy L from the challenger. At this time, if L does not satisfy W the challenger sends the secret key SK corresponding to L to the attacker. The attacker can repeat Phase 1 as needed.Challenge: The challenger runs the encrypt algorithm to derive $\left(<C_{0,\;}C_{1}>,\mathrm{Key}\right)$. The challenger sets ${\mathrm{Key}}_{0}$ = Key for the original Key, and then generates a random key ${\mathrm{Key}}_{1}$. At this time, the sizes of the two keys ${\mathrm{Key}}_{0},{\mathrm{Key}}_{1}$ are the same. The challenger selects a random value b∈{0,1} and sends $\left(<C_{0,\;}C_{1}>,{\mathrm{Key}}_{b}\right)$ to the attacker.Guess: The attacker guesses $b^{\prime }\in \left\{0,1\right\}$ for the ciphertext. If L does not satisfy W and $b^{\prime }=b$, then the attacker is defined as winning. The final definition of the probability that an attacker can win the game in this game is $\mathrm{Pr}\left[b^{\prime }-b\right]-1/2$. 

**Definition** **1.**
*The algorithm is termed a semantic security algorithm if the attacker receives a negligible benefit in the above game within a polynomial time t [25,26,27].*


## 3. Security Requirements

This section looks at security of the cloud-based IoMT environment, and describes the problems that may occur when using CP-ABE as shown in Figure 5.
Collusion attacks: Users can infer other users’ attributes through collusion with each other and generate another user’s secret key with the inferred attributes. Therefore, when a AA generates a secret key, it is necessary to generate a secret key by applying various variables in addition to the user’s attributes. In addition, users can leak data through collusion attack with service providers. Therefore, even if data is leaked through a collusion attack, data security technology so only a legitimate user can decrypt and view it is required.**Unauthorized user access control:** Since the cloud is a public environment with a large scope, anyone can access the data stored therein, thereby creating various security threats. Accordingly, access control technology and security technology for accessing stored data are required. If attribute-based encryption is used among security technologies, only users with attributes previously specified by the data owner can access the stored data. Therefore, it is necessary to apply attribute-based encryption because it can provide the confidentiality and integrity of data.Tracking users through a distributed key: The problem of the basic CP-ABE method is that there is no value that can identify the user issued a key, so it is impossible to identify the user who was first issued a distributed key. Therefore, if the distributed key is misused, study is needed to verify the identity of the user first issued the key through a tracking process [28].**User privacy protection:** Attribute-based encryption ensures anonymity because data owners and users encrypt and decrypt data with their own attributes. However, in order to provide traceability, the user’s privacy can be infringed by exposing the user’s identifier value to the AA when the key is issued by the attribute verification agency. Therefore, research is needed to protect users’ privacy in the cloud.**Verify data integrity as uploaded by data owner:** In existing CP-ABE schemes, it is assumed that if the user accesses the ciphertext uploaded by the data owner and decrypts the ciphertext to obtain the message, it is a legitimate message. In addition, it is assumed that the result of the partially decrypted ciphertext is a legitimate message because the outsourcing server is trusted in the schemes. However these are in correct assumptions. The message uploaded to the cloud can be falsified, and it is not known whether the value calculated by the outsourcing server is the correct value [29,30,31]. Accordingly, it is necessary to verify whether the user’s final decrypted value is the original message of the data owner.**Efficiency:** In some of the existing CP-ABE methods, the size of the ciphertext is proportional to the number of attributes specified in the access structure when generating the ciphertext. Accordingly, the size of the ciphertext increases linearly with the number of attributes, which occupies costly cloud storage space. In addition, the size of the ciphertext increases the burden of computation for the decrypting user. In order to solve this, it is necessary to introduce a server that supports outsourcing that can partially process the computation amount, and it is necessary to study how to reduce the computation amount and reduce the burden.

## 4. Proposed Scheme

This section proposes a CP-ABE-based access control method that provides user privacy protection and key abuse prevention for secure data sharing in a medical cloud environment.

As shown in Figure 6, the participants in this proposed scheme consist of five objects: data owner, cloud server (storage, access control (AC)), tracing authority (TA), attribute authority (AA), and user. A detailed description of the object role and protocol flow proposed in this paper are below.

### 4.1. System Model

#### 4.1.1. System Objects

The roles of each object in this proposed scheme are as follows.
**Data Owner (Patient):** Data Owner is a user who uses the cloud to store encrypted data. They create an access structure based on the attributes of the users who can access their data, encrypt the data, and upload it to the cloud.**Medical Cloud Server:** Cloud servers consist largely of storage and access control (AC). Storage is where encrypted data is stored, and an AC server is a trusted server that supports outsourcing operations. The AC server role controls user access and processes part of the decryption operation after comparing the attributes of the access structure specified in the ciphertext with those of the user who requested the ciphertext. As a result, the decryption computations of the user is reduced, thereby increasing the efficiency of the user’s decryption operation.**Trace Authority (TA):** A trusted server that manages user information. The user registers with the TA before the key is issued by the AA. TA generates and issues the user’s anonymous ID value. In the event of a problem with a leaked key, the AA can then be used to track down and identify the first user to whom the key was issued.**Attribute Authority (AA):** As a semi-trusted server, it manages the user’s attributes and creates a key that can decrypt in CP-ABE. A key is generated and sent to the user that allows the user to decode the ciphertext based on the user’s attributes at the time of the key request. At this time, the AA generates a key with the user’s attributes, so no value exists to identify the user and therefore the user cannot be identified. It can later work with the TA to trace the user who was issued the key.**User (Doctor or Nurse, etc.):** Users access encryption data stored on cloud storage through their attributes. The data is obtained by performing partial decryption using their attributes and final decryption using keys issued from the AA.

#### 4.1.2. System Overall Scenario

This proposed scheme is a CP-ABE-based data sharing system in the medical cloud environment that provides user privacy protection and key abuse prevention, and ensures the integrity of the data uploaded by the data owner through a verification phase. Assuming the data owner as a patient, as shown in Figure 6, the patient creates an access structure with the attributes of the user who has access to his data, and uploads the data by encrypting it. Then, among data users (doctors, nurses, etc.) who can access the medical cloud server, only users corresponding to the attributes specified by the patient can check patient data. In the scenario of this paper, the sharing or delegation of credential (key) authority to access is not considered, and only users who have been granted access authority from AA can check the ciphertext uploaded by the data owner, assuming that the doctor has shared or delegated permission to access patient data (credential, key) with other users. In the future, problems such as leakage of patient’s personal information stored in the server from the user who received the delegated key or alteration of server data may occur. At this time, the characteristics of the thesis are that the identity can be verified by tracking the user who first shared or delegated the key, and the integrity of the data can be verified in the outsourced data processing stage. Data sharing is mainly performed in a six-phase scenario, and a tracking phase is performed to trace and identify the user who was first issued the key in the case of a leaked key. Each phase proceeds as follows.**User registration and key issuance phase**: The user registers with the TA before receiving the key from the AA. The TA registers its own unique identifier and ID values, creates an anonymous $ID_{i}$, and sends it to the user (shown in steps 1–2 in Figure 6).Setup(k): The public key PK and master key AMK are generated by the AA by inputting the security parameter k (shown in steps 3 in Figure 6). KeyGen(MK, S,PK): The user sends $ID_{i}$, and attributes to the AA. The AA uses the values received from the user to generate a secret key SK capable of decrypting the ciphertext and transmits SK to the user (shown in steps 3–4 in Figure 6).Encrypt(PK, M, AS): The user encrypts the message with a normal symmetric key. After that, the access structure (AS) is created, and the symmetric key that encrypts the message with the PK and AS is encrypted to create a ciphertext CS. The ciphertext $CT^{\prime }$ includes the CS that encrypted the message, the CT that encrypted the key, the access structure AS, and the message verification key value VK (shown in steps 7–8 in Figure 6).**User data access and decryption phase**: The user creates a token to access the cloud, the AC server authenticates the user and partially decrypts the encrypted text, and the user performs final decryption. The partial decrypt phase and the final decrypt phase are as follows (shown in steps 9–13 in Figure 6).Partial decrypt(CT, S)**:** The AC server performs partial decryption when satisfied that the use attribute set matches the attribute set contained in the ciphertext. After the partial decoding process, the result C and the ciphertext $CT^{\prime }$ are sent to the user.$Final\;decrypt\left(C,PK,SK,CT^{\prime }\right)$**:** The user performs the final decryption of C and $CT^{\prime }$ received from AC using secret key SK. If the decryption is done correctly, the user can get the key that encrypted the message. The user obtains the message by decrypting the ciphertext with the key and performs the verification step to verify the integrity of the message.**Tracking the user who first issued the key:** It is possible to verify the identity of the user who was first issued the key by tracing the distributed key. This can solve the key abuse problem (shown in steps 14 to 16 in Figure 6).

### 4.2. Proposed Schemes

#### 4.2.1. System Parameters

The system parameters of this proposed scheme are as follows.Storage: Servers that manage dataAccess Control (AC): User access control managementTrace Authority (TA): Trusted authority that manages user information and traces leaked keysAttribute Authority (AA): Authority that verifies user attributes and issues keysMSK: master keyPK, AMK: Public parameter and master key required for attribute-based encryptionSK: User security key (decryption key)UTR: Value for tracking users$R{\mathrm{ID}}_{i},{\;\mathrm{ID}}_{i,1}$: User real ID, user created ID${\mathrm{ID}}_{i}$: User anonymous ID$T_{i}$: Valid anonymous ID period$\left(T_{pub_{TA},\;}\beta \right)$: TA’s public and private key pairs$\left(T_{pub_{AA},\;}\alpha \right)$: AA’s public and private key pairs$A_{U},\;A$: User attribute data, A set of attribute dataAS: Access policyTK: Tokens for access to the cloudC: Partially decrypted ciphertextCT,CS: Data with encrypted key, data with encrypted message$CT^{\prime }$: Data with key and message encrypted ($CT^{\prime }=<CT,CS>$)T: Timestamp 

#### 4.2.2. Assumptions

The proposed scheme assumes the following.Initially, TA and AA define two large primes p,q and the elliptic curves as follows for the security parameters.
(5)$$y^{2}=x^{3}+ax+b\left(mod\;p\right),\;a,b\in F_{P}$$The public key and private key of TA can be represented by $T_{pub\_TA}=\beta \cdot P$, MSK={β}, and the public key and private key of AA are $T_{pub\_AA}=\alpha \cdot P,$
MSK={α}.The AA issues a key after verifying the user attributes. At this time, part of the user identity value ($ID_{i},{\;Q}_{ID_{i}}$) is shared with the AC server.

#### 4.2.3. User Registration and Key Issuance Phase

**Step 1.** Initially, the user requests registration by sending his unique identifier and ID value $\left(RID_{i},ID_{i,1}\right)$ to the TA as a registration message. The TA calculates $ID_{i,2}$ after checking the user’s unique identifier value $RID_{i}$. After that, the user anonymous identification value $ID_{i}$ is generated and transmitted to the user. Hash functions: $h_{1}$: {0, 1} ∗ →$Z_{q}^{*}$.
(6)$$ID_{i,2}=RID_{i}⊕h_{1}\left(\beta \cdot ID_{i,1},T_{i},T_{pub_{TA}}\right)$$
(7)$$ID_{i}=\left\{ID_{i,1},ID_{i,2},T_{i}\right\}$$

**Step 2.** User sends anonymous $ID_{i}$ and his attribute set $A_{U*}$ to AA. AA creates a public key master key for data owners and users through the setup process. Assume the following during the setup process: Attribute universal set is $U=\left[att_{1},\;att_{2},att_{3},\ldots att_{n}\right]$. Each attribute has a multi-value of $V_{i}=\left[v_{i,1,\;}v_{i,2,\;}v_{i,ni,}\right]$ and can be represented as $att_{i}$ as a set. $W=\left[W_{1},\;W_{2},W_{3},W_{n}\right]$ is an access policy, where $W_{i}\subset V_{i}$. When the prime order of bilinear group G is p, AA generates a random constructor $\alpha ,\beta ,\;t_{i}\in Z_{p}$. After calculating $Y=e{\left(g,g\right)}^{\alpha },\;h=g^{\beta }\in G_{0},\;T_{i}=g^{t_{i}}$, a public key and a master key are generated as shown.

<Setup=PK, AMK>
(8)$$PK=\left\{e,\;g,\;{\left\{T_{i}=g^{t_{i}}\right\}}_{i\in \left[1,n\right]},\;h=g^{\beta },\;e{\left(g,g\right)}^{\alpha }\right\}$$
(9)$$AMK=\left\{\alpha ,\;{\left\{t_{i}\right\}}_{i\in \left[1,n\right]}\right\}$$
(10)$$T_{pub\_AA}=\alpha \cdot P,\;MSK=\left\{\alpha \right\}$$

After that, the key SK, which uses anonymous $ID_{i}$ and attribute $A_{U*}$ transmitted from the user is generated through the Keygen process. The steps for generating the secret key SK are as follows.

<$KeyGen\left(PK,ID_{i},\;AMK,A_{U*}\right)=SK$>-$random\;number\;d_{i}\in Z_{q}{}^{*}$, $Q_{ID_{i}}=d_{i}P$-j∈S (The symbol indicates the number of each attribute as j, and the set of attributes is expressed as S)-$r_{1}\ldots r_{j}\in Z_{p}$ (The random value given for each property is represented by the symbol $r_{j}$.)-hash functions: $h_{2}$: {0, 1}* → ${\left\{0,\;1\right\}}^{l_{m}}$, where $l_{m}$ denotes the bit length of the messages.
(11)$$PSK_{ID_{i}}=d_{i}+h_{2}\left(ID_{i},Q_{ID_{i}}\right)\cdot \alpha$$
(12)$$UTR=ID_{i}⊕h_{2}\left(ID_{i},Q_{ID_{i}}\right)$$
(13)$$r=\overset{n}{\underset{j=1}{\sum }}r_{n}$$

Then, calculate $D^{\prime }=g^{\alpha -r}$. The secret key generation is performed as follows.
(14)$$SK=\left\{S,PSK_{ID_{i}},UTR,\;D^{\prime }=g^{\alpha -r},{\left\{D_{i,1}=g^{r_{i}}\right\}}_{i\in \left[1,n\right]}\right\}$$

After issuing the secret key SK to the user, the user information ($ID_{i},{\;Q}_{ID_{i}}$) is shared with the AC server (shown in Figure 7).

#### 4.2.4. Data Encryption Phase

In this phase, the data owner encrypts the data and transmits it to the cloud storage for upload (shown in Figure 8). 

**Step 1**. Data owner creates access structures from the user’s attributes. Then, multi-values of the attributes contained in the access structure are then calculated according to the conditions, generating a ciphertext CT. Ciphertext $CT^{\prime }$ is composed of ciphertext CT that encrypts the KEY that can decrypt the message, a CS that encrypts the message, and the verification value VK for the KEY. By calculating the computed values for each attribute in the ciphertext CT, the size of the ciphertext can be reduced, and thus the space of the previously wasted cloud storage can be efficiently used.

<$Encrypt(\mathrm{PK},\;M,\;\mathrm{AS})=CT^{\prime }$> -$M\in G_{T}$ and access policy $W=\left[W_{1},\;W_{2},W_{3},W_{n}\right].$-Random value generation $s\in Z_{p}\;\mathrm{such}\;\mathrm{that}$
(15)$$s=\overset{n}{\underset{j=1}{\sum }}s_{j}$$

Then, it is calculated as follows.
(16)$$\tilde{C}:Y^{s}=\mathrm{KEY}\cdot e{\left(g,g\right)}^{\alpha s},\;\overset{ˇ}{C}:g^{s},\;\bar{C}:h^{s},$$

If $v_{i,1\;}\in W_{i}$, computes
(17)$$C_{i}=g^{t_{n}s}$$

If $v_{i,1\;}\notin W_{i}$, computes
(18)$$C_{i}=g^{t_{n+1}s}$$
(19)$$\mathrm{CT}=<\tilde{C},\;\overset{ˇ}{C},\bar{C},\;C^{\prime }>$$
(20)$$\;C^{\prime }={\left(h\cdot \prod_{i\in \mathrm{AS}}C_{i}\right)}^{s}\;=\;{\left(h\cdot \prod_{i\in \mathrm{AS}}g^{t_{i}}\right)}^{s},{\;g}^{t}=\overset{n}{\underset{i=1}{\prod }}g^{t_{i}}$$
(21)$$\mathrm{CT}=<\mathrm{AS},\tilde{C},\;\overset{ˇ}{C},\;\bar{C},\;C^{\prime }>$$
(22)$$\mathrm{CS}\;={\;\mathrm{Enc}}_{\mathrm{KEY}}\left(M\right),\;\mathrm{VK}=\left(g^{h\left(\mathrm{KEY}\right)},g^{h\left(M\right)}\right)$$
(23)$${\mathrm{CT}}^{\prime }\;=\;<\mathrm{CT}\;,\;\mathrm{CS},\;\mathrm{VK}>$$

The cloud storage safely stores the ciphertext CT received from the user.

#### 4.2.5. User Data Access and Data Decryption Phase

In this phase, the user accesses the cloud, receives the ciphertext, and then decrypts it. The decryption step consists of two steps: partial decoding on the AC server and final decoding by the user (shown in Figure 9).

**Step 1.** The user creates a TK that can access the cloud through the $PSK_{ID_{i}}$ from the secret key SK received from AA. Then, when requesting access to the cloud via $ID_{i}\left|\left|TK\right|\right|T$, the AC server verifies ($ID_{i},{\;Q}_{ID_{i}}$) of the TK value through the $PSK_{ID_{i}}$ user information values shared by AA and determines that it is a legitimate user.
(24)$$\mathrm{Token}\;\left(\mathrm{TK}\right)=PSK_{ID_{i}}||A_{U*}$$

Since the AC server knows the value of ($ID_{i},{\;Q}_{ID_{i}}$) for the user, it verifies the user by performing the following verification process
(25)$$PSK_{ID_{i}}\cdot P=Q_{ID_{i}}+h_{2}\left(ID_{i},Q_{ID_{i}}\right)\cdot T_{pub\_AA}$$

When it is verified that the user is a legitimate user, a partial decryption is performed after requesting the ciphertext desired by the user from the storage and comparing the access structure and user attribute values specified in the ciphertext.

<$Partial\;decrypt\left(\mathrm{CT},\;A_{U*},x\right)=C>$

The process of performing partial decryption with ciphertext and user attributes is as follows.
(26)$$C=\frac{e\left(g^{r},C^{\prime }\right)}{e\left(\overset{ˇ}{C},\;{\left(\prod_{j\in S}g^{t_{i}}\right)}^{s}\right)}=e{\left(g,g\right)}^{rs}$$

**Step 2.** The AC server transmits the result C and the ciphertext $CT^{\prime }$ obtained by performing partial decryption to the user.

**Step 3.** The user performs the final decryption of the $CT^{\prime }$ received from the AC server with SK, C, and PK, and extracts a key that can decrypt the message. Thereafter, after verifying whether the existing message can be decrypted, the ciphertext CS is finally decrypted.

<$Final\;decrypt\left(\;\mathrm{PK},\;\mathrm{SK},C,\;CT^{\prime }\right)=M$> 

The user performs the final decryption to obtain a key capable of acquiring the message M.
(27)$$KEY^{\prime }=\frac{\tilde{C}}{e\left(\overset{ˇ}{C},D^{\prime }\right)\cdot C}=\frac{\mathrm{KEY}\cdot e{\left(g,g\right)}^{\alpha s}}{e\left(g^{s},g^{\alpha -r}\right)\cdot e{\left(g,g\right)}^{rs}}$$
(28)$$M^{\prime }\;=\;Dec_{KEY^{\prime }}\left(CS\right)$$

Create ${\mathrm{VK}}^{\prime }$ with $M^{\prime }$ and $KEY^{\prime }$. After that, the verification process is performed by comparison to the existing VK and the integrity of the message is confirmed.
(29)$$\mathrm{VK}={\mathrm{VK}}^{\prime }=\left(g^{h\left(\mathrm{KEY}\right)},g^{h\left(M\right)}\right)=\;g^{h\left(KEY^{\prime }\right)},g^{h\left(M^{\prime }\right)}$$

#### 4.2.6. Tracing Phase

This step is intended to prevent unauthorized third parties from accessing the cloud by receiving a key from someone. At this time, the investigating agency cannot know the process of distributing the secret key SK, but it can know the identity of the user who received and distributed the key for the first time through the tracking process. Here, we assume that $SK=\{S,PSK_{ID_{i}},UTR,\;D^{\prime }=g^{\alpha -r},{\left\{D_{i,1}=g^{r_{i}}\right\}}_{i\in \left[1,n\right]}\}$ has been distributed and we proceed to the first step (shown in Figure 10).

**Step 1.** AA verifies $PSK_{ID_{i}}$ in the parameters of the distributed key and extracts the user’s anonymous ID value $ID_{i}$. Thereafter, the user’s anonymous ID value $ID_{i}$ is transmitted to the tracing authority agency TA.
(30)$$PSK_{ID_{i}}\cdot P=Q_{ID_{i}}+h_{2}\left(ID_{i},Q_{ID_{i}}\right)\cdot T_{pub\_AA}\;\left(\mathrm{Verification}\;\mathrm{complete}\right)$$
(31)$$ID_{i}=UTR⊕PSK_{ID_{i}}$$

**Step 2.** The TA extracts the corresponding unique ID value $RID_{i}$ after receiving the user’s anonymous ID value $ID_{i}$. Afterwards, the identity of the user who issued the key for $RID_{i}$ is confirmed.
(32)$$ID_{i}=\left\{ID_{i,1},ID_{i,2},T_{i}\right\}$$
(33)$$RID_{i}=ID_{i,2}⊕h_{1}\left(\beta \cdot ID_{i,1},T_{i},T_{pub\_TA}\right)$$

## 5. Analysis of Proposed Scheme

This proposed scheme satisfies the security requirements give in Section 3. In Section 5.1, the security requirements of this proposed method were analyzed, and in Section 5.2, the efficiency of operation was analyzed (shown in Table 3 and Table 4).

### 5.1. Security Analysis

**Collusion attack:** In the proposed scheme, AA uses nonce value in addition to the attribute values when generating the secret key, so even if the user determined the attributes through a collusion attack, the secret key can be generated. In addition, in the part where the partial decryption is performed by comparing the attributes of the user and the attributes of the access policy specified in the ciphertext on the AC server, the message M cannot be viewed because the user’s secret key is not known. Therefore, it is safe from collusion attack between users or between users and service providers.**Authenticated user access control:** In the proposed scheme, only the user who generated the token by receiving the secret key from AA can access the data. In addition, only users who satisfy the attributes of the access structure specified by the data owner can access the data stored in the cloud. The user’s access is primarily blocked by the AC server, and when the above conditions are satisfied partial decryption is performed and the ciphertext and the result of partial decryption are transmitted to the user. Therefore, the proposed scheme provides confidentiality and integrity to stored data because only authenticated users can access it.**Tracing the user first issued an abused key:** The existing scheme for providing traceability shown in Table 1 verify the leaked key via a Key Sanity Check, and after verification, the user’s identification information contained in the key is verified and the AA identifies the user who was issued the key for the first time. Our proposed scheme also includes a value $PSK_{ID_{i}},UTR$ that can identify a user from SK. The difference with the schemes presented in Table 1 is the aspect of user privacy protection. Existing schemes manage the user’s information in AA because the key must include information that can identify the user when issuing the key. This can violate the user’s privacy, excluding the user’s anonymity, which is provided by default in attribute-based encryption. Therefore, in this proposed scheme, an entity that issues an anonymous ID value called a TA is provided, and the user is provided with an anonymous ID value when registering so that the CP-ABE data sharing system can be utilized anonymously. In the event a key is misused, the AA and TA can cooperate with each other to track and identify the user who was first issued the key, and user privacy can be protected.**Verify data integrity:** In our proposed scheme, the attribute-based encryption/decryption is not the message but the message encryption key, unlike existing schemes. Therefore, in the partial decoding process in the outsourcing server, the message is not converted. In addition, in the proposed ciphertext $CT^{\prime }$ = <CT, CS, VK>, there is a verification key VK that can verify the integrity of the message of the data owner. Accordingly, the user verifies through the $KEY^{\prime }$ and M′ obtained after the final decoding $\mathrm{VK}={\mathrm{VK}}^{\prime }=\left(g^{h\left(\mathrm{KEY}\right)},g^{h\left(M\right)}\right)=\;g^{h\left(KEY^{\prime }\right)},g^{h\left(M^{\prime }\right)}$ to verify the integrity of the message. Compared to the existing Yu and Jiguo Li schemes that support verifiable outsourcing, the verification of message integrity is relatively simple. 

### 5.2. Efficiency

The throughput experiments in Figure 11 and Figure 12 were performed on a Windows system with a 3.50 GHz Intel Core i5-4690 processor and 8 GB of RAM. For the pairing operation, see Pairing Based Crypto Library (Lynn, B., “The pairing-based cryptography (PBC) library,” Available: http://crypto.stanford.edu/pbc, 2012.). The symmetric key cipher used for comparison measurement is AES-128, and the parameter of bit operation is 128 bits. When performing encryption/decryption once with a symmetric key, the message length was based on 1000. In addition, because the operation speed is very small for multiplication operations and exponentiation operations, it is expressed as 0.001. In Figure 11 and Figure 12, the values are expressed on a millisecond basis.**Cloud storage space efficiency:** In existing CP-ABE methods, when the ciphertext is generated, the size of the ciphertext increases in proportion to the number of attributes specified in the access policy, thus wasting storage space. In particular, in the Qi Li scheme, the size of the ciphertext $\mathrm{CT}<T,\;C,\;C_{1},C_{2},{\left\{C_{i,1},C_{i,2},C_{i,3}\right\}}_{i\in \left[m\right]}>$ increases with the number of attributes. In the proposed scheme, the number of attributes specified in the access policy is represented as a single number by performing a separate operation $C^{\prime }={\left(h\cdot \prod_{i\in AS}C_{i}\right)}^{s}$ = ${\left(h\cdot \prod_{i\in AS}g^{t_{i}}\right)}^{s}$, which results in a constant-size ciphertext. As a result, as opposed to existing CP-ABE methods, the proposed scheme generates a ciphertext of a certain size, not proportional to the number of attributes specified when generating the ciphertext. The wasted cloud storage space can be used efficiently. However, only the size of the ciphertext is constant. The amount of computation required for encryption varies depending on the number of attributes.**Efficiency of computation:** In conventional CP-ABE methods, when decrypting the user receives the ciphertext and decrypts it. Accordingly, the amount of computation for the use is proportional to the size of the ciphertext. This proposed scheme performs partial decoding by including an AC server that supports outsourcing. As the result of partial decryption by the AC server, the user receives the C and the ciphertext $CT^{\prime }$ and proceeds to the final decryption, so the message M can be obtained. By using the AC server for outsourcing, the computational efficiency can be increased by reducing the amount of computation required for the user. As shown in Table 4, this proposed scheme has more user decryption computations than the Qi Li and Premkamal, Jiguo Li schemes among the CP-ABE methods that support an outsourcing server. This is because most of the decryption operations are performed on the server, so the more users there are, the more computations the server processes. However, in the Premkamal scheme and Jiguo Li scheme, when the user distributes the key to someone, the distributed key can be used for access by unauthorized users. At this time, if the distributed key is misused, the user who originally issued the key cannot be tracked. In other words, the aforementioned key abuse problem arises. Traceability is provided in the qi Li system, but personal privacy issues may arise because the server that issued the key knows the user’s information. Our proposed scheme solves the problem of key abuse by tracking the user who was issued the key for the first time through the leaked key and provides user anonymity.

## 6. Conclusions

This proposed scheme is a CP-ABE-based medical data sharing system that supports major exploit prevention and outsourcing operations and allows medical data to be safely and efficiently shared in the cloud in IoMT environments. It provides traceability by tracking the user who was issued a key through the distributed key, but preserves the user’s anonymity because the key is issued by the AA using the user’s anonymous ID value. In addition, it is safe against various security threats such as masquerade attacks and collusion attacks, and unauthorized third party access is not possible. In terms of efficiency, the ciphertext size of the existing CP-ABE scheme was proportional to the number of attributes. However, this proposed approach effectively removes the wasted space of existing cloud storage because the size of the ciphertext is not proportional to the number of attributes, like $\;C^{\prime }={\left(h\cdot \prod_{i\in AS}C_{i}\right)}^{s}$ = ${\left(h\cdot \prod_{i\in AS}g^{t_{i}}\right)}^{s}$. In addition, the calculation efficiency is higher than the Hahn scheme, Jiang scheme, and Yu scheme CP-ABE methods, which do not provide outsourcing of existing ciphertext and outputs of a certain size in the process of user decrypting ciphertext. The message verification operation is also 2H, which is more efficient than other methods. Lastly, when decrypting a ciphertext on the user’s side, since some of the decryption operations are supported by the AC server, the computational efficiency for users who are burdened with the decryption process may be increased due to lack of computing resources.

In the future research, this proposed scheme has a lot of computation to verify the user’s identity through the key compared to the existing schemes. Therefore, a lightweight CP-ABE scheme is needed in order to increase the efficiency of computation. Additionally a key escrow problem may occur in the AA, so research is needed to solve it. 

## Figures and Tables

**Figure 1 sensors-20-04934-f001:**
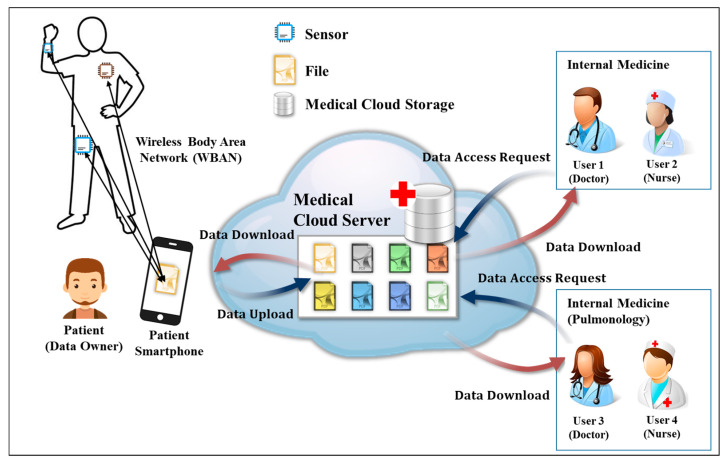
Data sharing system in a medical cloud environment.

**Figure 2 sensors-20-04934-f002:**
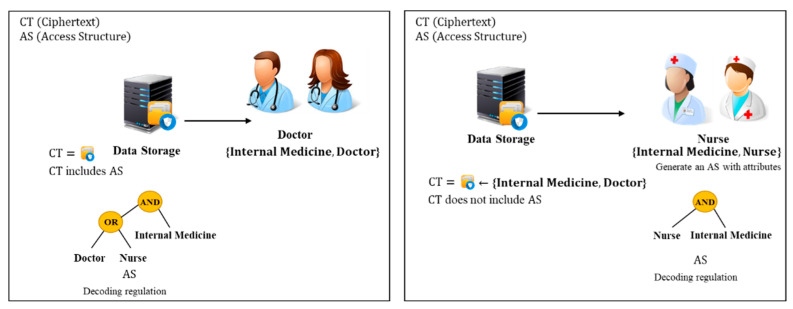
Attribute-Based Encryption (Left ciphertext-policy attribute-based encryption (CP-ABE), Right key-policy attribute-based encryption (KP-ABE)).

**Figure 3 sensors-20-04934-f003:**
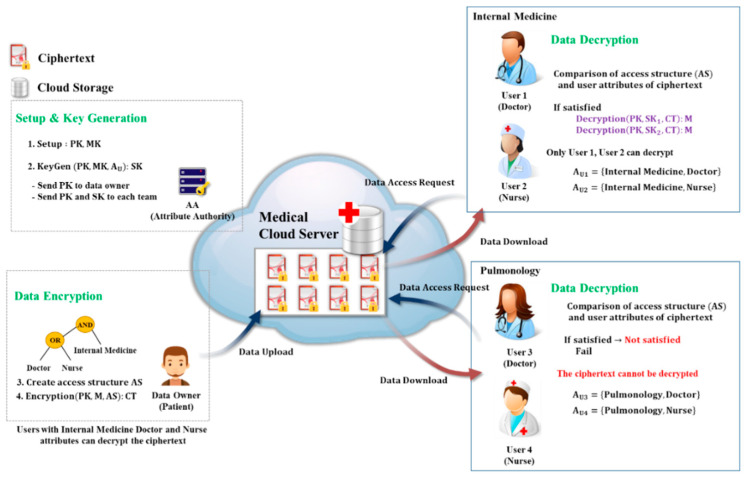
Medical data sharing system based on CP-ABE in a cloud Internet of Medical Things (IoMT) environment.

**Figure 4 sensors-20-04934-f004:**
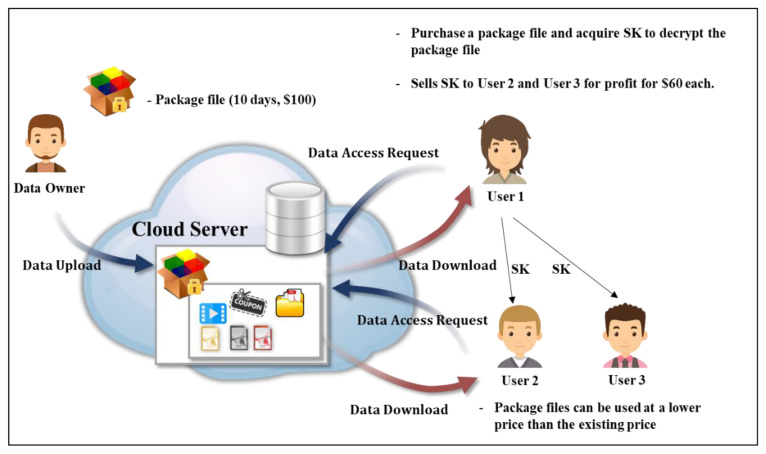
Key abuse and misuse issues that can occur in the CP-ABE data sharing system.

**Figure 5 sensors-20-04934-f005:**
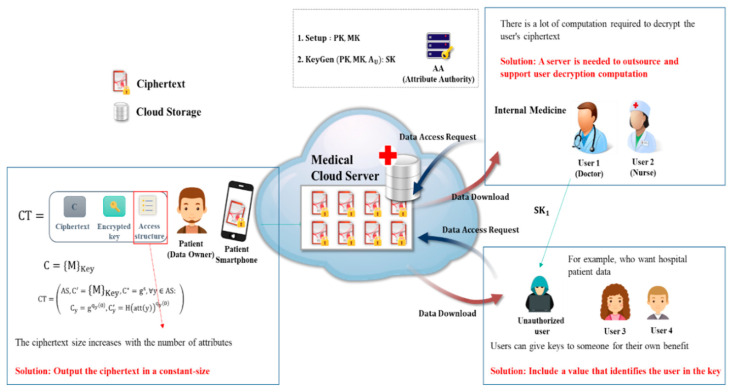
Problems that can occur in CP-ABE.

**Figure 6 sensors-20-04934-f006:**
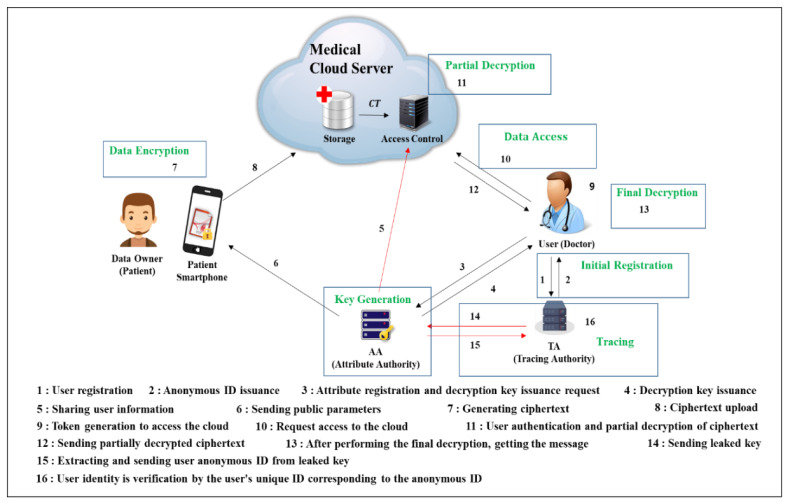
Overall scenario of the proposed scheme.

**Figure 7 sensors-20-04934-f007:**
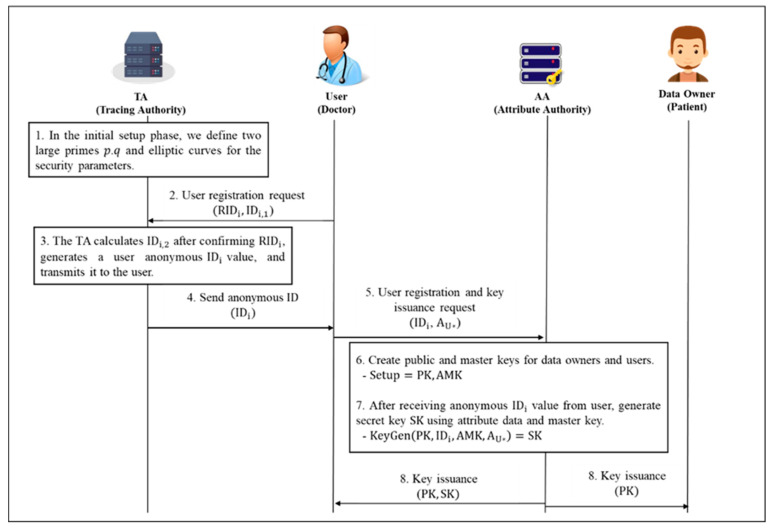
User Registration and Key Issuance Phase.

**Figure 8 sensors-20-04934-f008:**
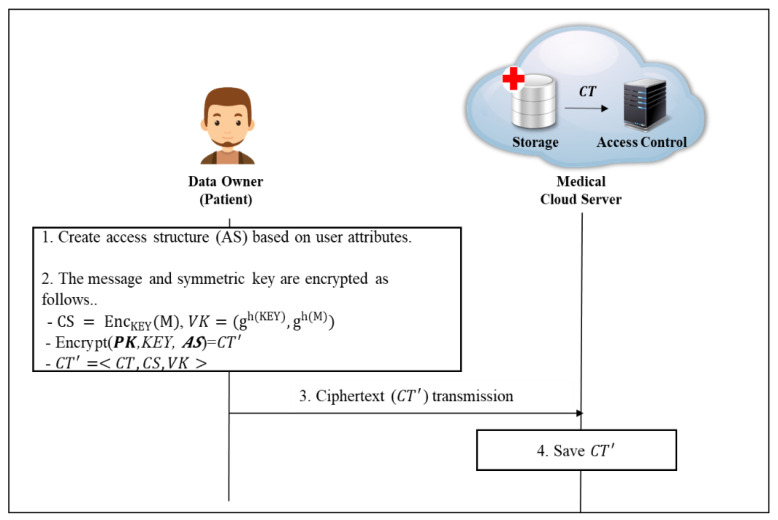
Data Encryption Phase.

**Figure 9 sensors-20-04934-f009:**
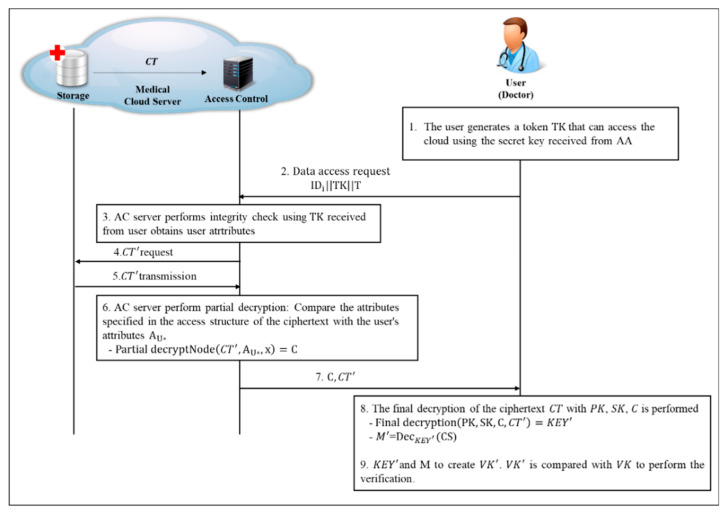
User Data Access and Data Decryption Phase.

**Figure 10 sensors-20-04934-f010:**
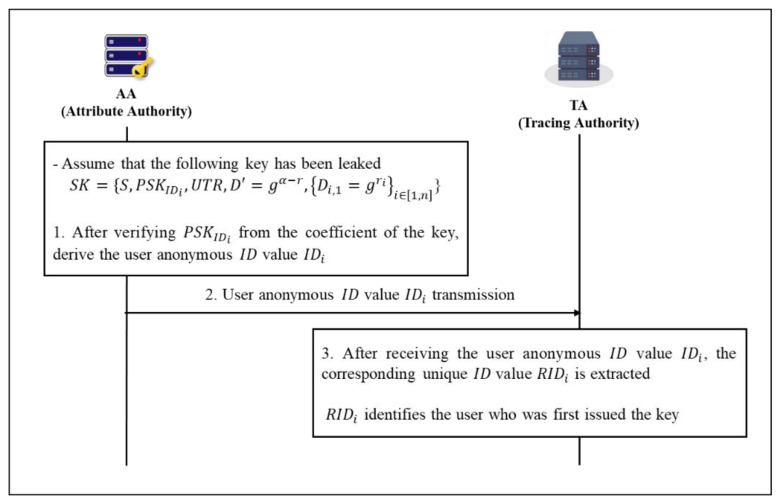
Tracing the User Who First Issued the Key Phase.

**Figure 11 sensors-20-04934-f011:**
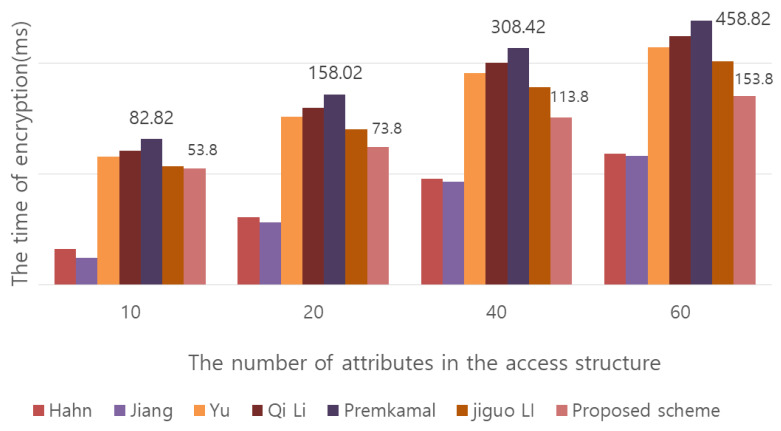
Comparison of data encryption time between existing CP-ABE schemes and the proposed scheme.

**Figure 12 sensors-20-04934-f012:**
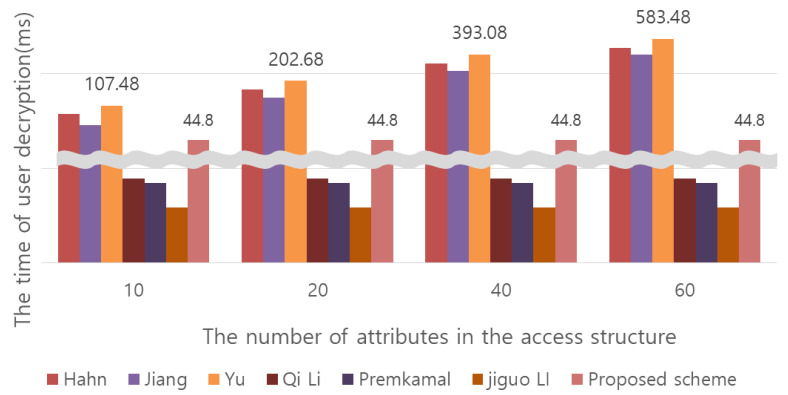
Comparison of data decryption time between existing CP-ABE schemes and the proposed scheme.

**Table 1 sensors-20-04934-t001:** Comparison of CP-ABE schemes provided for traceability.

CP-ABE Scheme	User Privacy	Traceability	Ciphertext Size	Support on Outsourcing Server	Outsourcing Results Verification
Qi Li scheme [5]	Not protected	Provided using identity table on management server	Proportional to the number of attributes	Provided	Not provided
Liu scheme [6]				Not provided	
Hahn scheme [7]			Constant size ciphertext	Provided	
Yu scheme [8]		Provided through the signature value included in the key	Proportional to the number of attributes	Provided	
Luo scheme [11]				Not provided	
Jiang scheme [12]		Provided using identity table on management server			

**Table 2 sensors-20-04934-t002:** Comparison of CP-ABE schemes with outsourced verification.

CP-ABE Scheme	Ciphertext Size	Support on Outsourcing Server	Outsourcing Results Verification
Hahn scheme [13]	constant-size ciphertext	Provided	Failed to provide
Wei Teng scheme [14]		Failed to provide	
Helil scheme [15]	Proportional to the number of attributes	Provided	Failed to provide
Lai scheme [16]			Verify by inserting the MAC authentication code in the cipher text.
Premkamal scheme [17]			Message verification with VK, which can verify ciphertext.
Qin scheme [18]	constant-size ciphertext		The hash value is used to verify the accuracy of outsourcing decrypt.
Jiguo Li scheme [20]			Verify by inserting the MAC authentication code in the cipher text.
Zhidan Li scheme [21]	Proportional to the number of attributes		After partial decoding on 2 servers, verify that the results match.

**Table 3 sensors-20-04934-t003:** Comparison of security requirements of this proposed scheme and the existing CP-ABE.

	Hahn Scheme [12]	Jiang Scheme [12]	Yu Scheme [22]	Qi Li Scheme [5]	Premkamal Scheme [17]	Jiguo Li Scheme [18]	Proposed Scheme
Collusion/ masquerade attack	Safe						
Data Storage space	Efficient	Inefficient				Efficient	
Ciphertext length	Constant size	Proportional to the number of attributes				Constant size	
User privacy	Infringement impossible	Infringement Possible (Trusted servers manage user information through identity tables)			Infringement impossible		
Tracing the first user to distribute the key	Untraceable	Traceable via identity table	Traceable with user signatures stored in identity tables	Traceable via identity table	Untraceable		TA and AA work together to track who was issued the first key
Verifying the integrity of the data owner’s message	Not considered			Message verification with VK, which can verify ciphertext		Verify by inserting MAC authentication code in ciphertext	Verify by inserting the message hash value in the ciphertext
Outsourcing operation support	Supported	Not supported		Supported			

**Table 4 sensors-20-04934-t004:** Comparing the computation amount of this proposed scheme with existing CP-ABE schemes.

	Hahn Scheme [13]	Jiang Scheme [12]		Yu Scheme [22]		Qi Li Scheme [5]
Encryption	$c_{e}+\left(n+4\right)M$	$c_{e}+M+\left(n+1\right)E$		$2c_{e}+\left(n+3\right)M$ +(4n+6)E		$c_{e}+\left(4n+5\right)E+\left(2n+2\right)M$ +2H+1Enc
Partial decryption (server)	$2c_{e}+3E+2\mathrm{nM}$	-		-		$4{\mathrm{nc}}_{e}+\left(2n+3\right)M+\mathrm{nE}$
Final decryption (user)	$\left(2n+3\right)c_{e}+\left(2n+3\right)M$	$2{\mathrm{nc}}_{e}+\left(n+1\right)M$		$\left(2n+3\right)c_{e}+\left(2n+3\right)M+\left(2n+1\right)E$		E+M+H+1Dec
Message verification	-	-		-		2H
	Premkamal Scheme [17]		Jiguo Li Scheme [18]		Proposed Scheme	
Encryption	$2\left(n+1\right)c_{e}+\left(2n+1\right)E$ +H+M+1Enc		$2c_{e}+2\left(n+1\right)M$ +2(n+4)E+2H		$c_{e}+\left(n+1\right)M+\left(n+5\right)E$ +2H+1Enc	
Partial decryption (server)	$\left(n+3\right)c_{e}+\mathrm{nE}$		$4c_{e}$+2M		$2c_{e}+\mathrm{nM}+2E$	
Final decryption (user)	1Dec		2H+4E+4M		$c_{e}+2M+2H+1\mathrm{Dec}$	
Message verification	E+3M+H		2H+2E+2M		2H	
$c_{e}:\;Pairing\;operation;\;M:\;Multiplication\;operation;\;n:\;Number\;of\;attributes;\;E:\;Exponentiation\;operation;$ H: Hash function Enc: Symmetric key encryption; Dec: Symmetric key decryption						

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
