# Peer review of "A Study on CP-ABE-Based Medical Data Sharing System with Key Abuse Prevention and Verifiable Outsourcing in the IoMT Environment"

_sensors, 2020, doi:10.3390/s20174934_

Round 1
Reviewer 1 Report
The paper addresses a very relevant topic. Among others, it answers a very interesting question: how to identify when access keys are abused by non-authorized third parties when otherwise outsourcing of access keys are a feature of the usage context - in this case between doctors and patients.
The scheme, the scientific background, and the novelty are well explained. The necessity of verifiable outsourcing techniques is well-vocalized in many parts of the paper. (Is line 227 correct? verifiably -> verifiable?)
One note though: while getting into the problem context, the reader has the instant idea that the doctor, instead of sharing their access credentials, should provide an authorized token or key to the patient, and the doctor should be prevented any ways possible to share - or outsource - his/her own credentials, tokens or keys. This seems trivial; and somehow this is what becomes the solution as well, in the article - of course in a much more scientifically sound and mathematically proven manner. Nevertheless, the problem statement and the solution can - and should be - described in a brief, commonly understandable way in order to get the idea - and the solution - flying. The fact that this is the actual proposed solution starts to getting revealed around page 13, in section 4.2.3.
Actually, there are two Figure 7's - this should be corrected, and none of them (neither Figure 8) is mentioned in the text.
Minor notes:
- using the word "research" should be changed to some synonym, such as "investigation" in line 71, and 310, at least.
- line 46: in this figure of speech it sounds like the quality of life of the medical staff will be improved. It should be rephrased a little.
- line 104 - Each chapter of this paper... should be rephrased.
- The usage of the double turnstile symbol in section 2.3 should be elaborated.
Reviewer 2 Report
This manuscript analyses the access control on cloud services using CP-ABE.
- Please, add parity on the roles and the writting. You represent the doctor always as a man while the nurse a woman.
- Improve the english there are too many mistakes
- Table 3. 3rd row, 2nd and 3rd column the text is repeated. You can merge most of the columns.
- Add international approaches (publications outside China) on your research as the results.
- Add numerical results in Abstract and Conclusions
- Improve plots 10 and 11. Text size as the body.
- You need to perform more results. I doubt after 20-pages manuscript you can only extract two plots of results.
Round 2
Reviewer 2 Report
The manuscript has improved considerably by making the proposed changes, hence, no more changes are requested.